# Recent Trends of Sunscreen Cosmetic: An Update Review

**Le Thi Nhu Ngoc [1], Vinh Van Tran [1], Ju-Young Moon [2], Minhe Chae [3], Duckshin Park [4],\* and Young-Chul Lee [1],\***

[1] Department of BioNano Technology, Gachon University, 1342 Seongnam-Daero, Sujeong-Gu, Seongnam-Si 13120, Gyeonggi-do, Korea; nhungocle92@gmail.com (L.T.N.N.); vanvinhkhmtk30@gmail.com (V.V.T.)

[2] Department of Beauty Design Management, Hansung University, 116 Samseongyoro-16gil, Seoul 02876, Korea; bora7033@naver.com

[3] Biocell Korea Co., Ltd., 1-2F Janghan B/D,54 Bongeunsa-ro 30-gil, Gangnam-gu, Seoul 04631, Korea; christinechae7@gmail.com

[4] Korea Railroad Research Institute (KRRI), 176 Cheoldobakmulkwan-ro, Uiwang-si 16105, Gyeonggi-do, Korea

\* Correspondence: dspark@krri.re.kr (D.P.); dreamdbs@gachon.ac.kr (Y.-C.L.); Tel.: +82-10-3343-2862 (D.P.); +82-31-750-8751 (Y.-C.L.); Fax: +82-31-460-5367 (D.P.); +82-31-750-4748 (Y.-C.L.)

**Abstract:** Ultraviolet (UV) radiation has been demonstrated to cause skin disorders, including sunburn and relative symptoms of prolonged exposure. It has been reported that sunscreens have beneficial effects in reducing the incidence of skin disorders (sunburn, skin aging, and immunosuppression) through their ability to absorb, reflect, and scatter UV. Many commercial products have recently been manufactured from not only usual organic and inorganic UV filters, but also hybrid and botanical ingredients using typical formulations (emulsion, gel, aerosol, and stick). Particularly, these products have been supplemented with several preeminent properties to protect against the negative effects of not only UVB, but also UVA. However, the use of sunscreen has faced many challenges, including inducing photoallergic dermatitis, environment pollution, and deficiency of vitamin D production. Therefore, consumers should efficiently apply suitable products to improve sun protection. as well as to avoid the side effects of sunscreen.

**Keywords:** sunscreen; organic/inorganic/hybrid/botanical agents; emulsion/gel/aerosol/stick formulations; antioxidants/anti-pollutants/defense-blue light and -IR

## 1. Introduction

Solar radiation reaching the terrestrial surface comprises ultraviolet (UV), visible light, and infrared (IR) rays [1]. The spectra of all electromagnetic radiation range from 100 nm to 1 mm, in which UV radiation has the shortest wavelength (200–400 nm) compared to visible light (400–740 nm) and IR (760–1,000,000 nm). UV radiation constitutes about 10% of the total light output of the sun [2]. The broad spectrum of UV radiation is subdivided into three recommended ranges (UVA, UVB, and UVC). Therein, UVA has the longest wavelength (320–400 nm) but the least energy photon, while UVB wavelength is in the middle span (280–320 nm) and UVC has the shortest wavelength (100–280 nm) but the highest energy [2]. It has been reported that moderate sun exposure offers a number of beneficial effects, including production vitamin D [3], antimicrobial activity [4], and improved cardiovascular health [5,6]. However, long-term exposure to UV rays is considered to be a potential risk of skin cancer and acute and chronic eye injuries (Figure 1) [7].

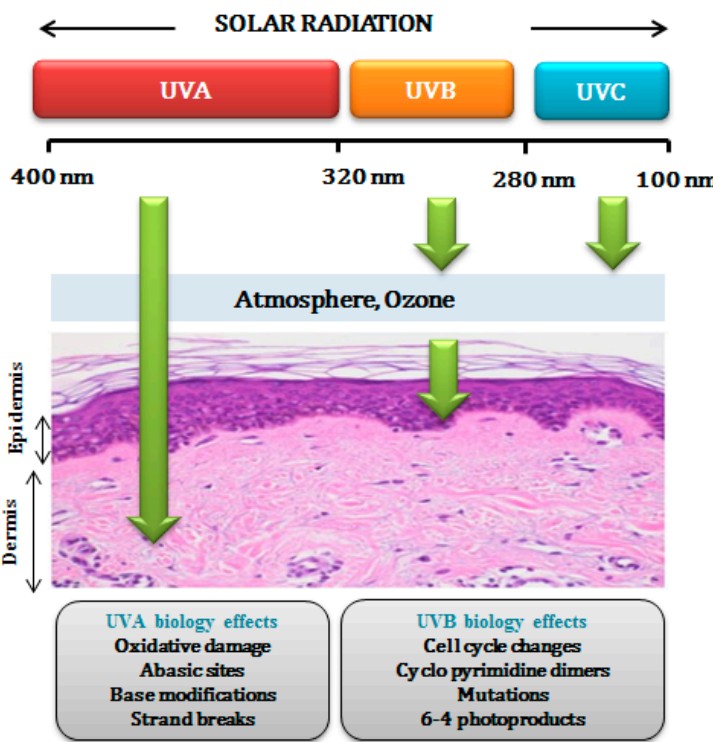

**Figure 1.** Electromagnetic spectra solar ultraviolet (UV) rays and their biologic effects on the skin [2].

UV-induced skin damage is one of the most common concerns in the world. Certainly, UVA is a risk of skin aging, dryness, dermatological photosensitivity, and skin cancer. It damages DNA through the generation of reactive oxygen species (ROS), which causes oxidative DNA base modifications and DNA strand breaks, resulting in mutation formation in mammalian cells [1,2,8]. On the other hand, UVB can directly damage DNA through the formation of pyrimidine dimer and then cause apoptosis or DNA replication errors, leading to mutation and cancer [5]. Although UVC is the shortest and most energetic wavelength, it is the most dangerous type of UV ray because it can cause various adverse effects (e.g., mutagenic and carcinogenic) [5]. However, UVC rays do not penetrate through the atmosphere layer.

It has been proved that photoprotectors, especially sunscreen, play a critical role in reducing the incidence of human skin disorders (pigment symptoms and skin aging) induced by UV rays [9]. Sunscreen was first commercialized in the United States in 1928 and has been expanded worldwide as an integral part of the photoprotection strategy [9]. It has been found to prevent and minimize the negative effects of UV light based on its ability to absorb, reflect, and scatter solar rays [10,11]. Over the decades of development, sunscreens have been improved step-by-step, accompanying the innovation of photoprotective agents [12]. Certainly, recent sunscreens are found to not only address UV effects, but also protect the skin from other risks (e.g., IR, blue light, and pollution) [13,14]. Indeed, while UV radiation is most commonly implicated in skin disorder development, it is crucial to note the potential role of these considerable harmful factors [13,15]. It has been suggested that these factors can worsen disorders of dyspigmentation, accelerating aging, and eliciting genetic mutations [15,16].

Furthermore, the photoprotective efficiency of sunscreen is determined through sun protection factor (SPF) and the protection grade of UVA (PA) values. According to Food and Drug Administration (FDA) regulations, commercial products must be labeled with SPF values that indicate how long they will protect the user from UV radiation and must show the effectiveness of protection [17]. Certainly, the SPF values are generally in the range of 6–10, 15–25, 30–50, and 50+, corresponding to low, medium, high, and very high protection, respectively [17]. Nevertheless, there are some fundamental misunderstandings of the SPF. Some argument is that an SPF 15 sunscreen can absorb 93% of the erythemogenic UV radiations, while an SPF 30 product can block 96%, which is just over

3% more (Figure 2) [18]. The argument may be correct when evaluating sun protection capacity, but is not sufficient in assessing the amount of UV radiation entering the skin. In other words, half as much UV radiation will penetrate into the skin when applying an SPF 30 product compared to an SPF 15 product. [18]. This is also illustrated by comparing SPF 10 with SPF 50 sunscreen. Ten and two photons transmit (%) through sunscreen film and enter the skin when applying SPF 10 and SPF 50 products, respectively, as a difference factor of five it is expected [18]. On the other hand, in 1996, the Japan Cosmetic Industry Association (JCIA) developed an in vivo persistent pigment darkening (PPD) method to evaluate UVA efficacy of sunscreen [19]. Sunscreens are labeled with PA+, PA++, PA+++, and PA++++, corresponding to the level of protection grade of UVA (PA) obtained from the PPD test [19,20]. Sunscreens labeled as PA+ express low protection, mainly contributed by between two and four UVA filters. Sunscreens containing four to eight sunscreen agents show moderate levels of UVA blocking and are labeled as PA++. In contrast, the PA+++ and PA++++ symbols represent products that are composed of more than eight UVA filters and provide a high sunscreen efficacy [10,19,20].

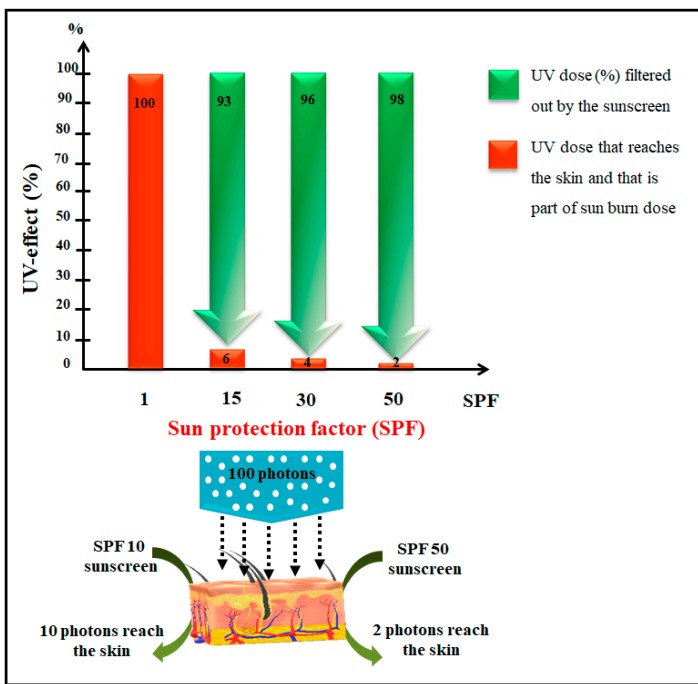

**Figure 2.** Illustration of the sun protection factor (SPF) definitions, including filtered and transmitted UV radiation [18].

In the literature, there are several published studies that fully introduce the basics of sunscreen products, such as ingredients, properties, and SPF evaluation, while formulas and novel properties (anti-pollutant, antioxidant, and blocking blue light and IR) have not clearly described. In order to provide a comprehensive summary of modern sun protection, this review specially focused on describing the ingredients and formulations of commercial products. In addition, it refers to the novel properties of sunscreen that can satisfy consumer demands, such as antioxidant, DNA repair enzymes, anti-pollutant, and defense-blue light and -IR.

## 2. Classification of Sunscreen Agents

Sunscreen agents are basically categorized into inorganic and organic UV filters which have specific mechanisms of action upon exposure to sunlight (Figure 3). Inorganic agents reflect and scatter light, while organic blockers absorb high-energy UV radiation [21,22]. Recently, hybrid materials combining properties of organic and inorganic compounds have attracted the attention of scientists as a promising sunscreen agent. Remarkably, botanical agents, which contain large amounts of

antioxidant compounds, can be used as inactive ingredients to protect the skin against adverse effects (e.g., photoaging, wrinkles, and pigment).

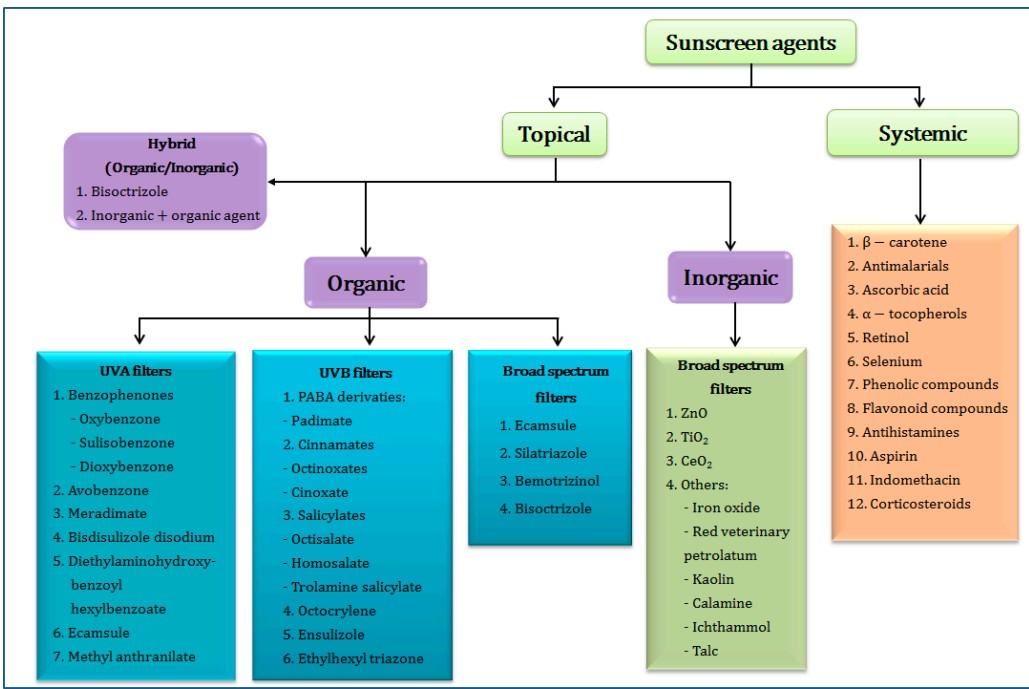

**Figure 3.** Sunscreen agent classification [23].

## 2.1. Organic UV Filters

Organic blockers are classified into either UVA (anthranilates, dibenzoylmethanes, and benzophenones) or UVB filters (salicylates, cinnamates, para-aminobenzoic acid (PABA)derivatives, and camphor derivative), which play an important role in absorption activity of sunscreen (Figure 3) [24]. These agents show outstanding safety and aesthetic properties, including stability, nonirritant, nonvolatile, non-photosensitizing, and non-staining to human skin, compared to inorganic UV filters [25]. Besides, they are mostly used in combination at levels currently allowed by the FDA to provide broad-spectrum absorption, as well as increased SPF values [26]. Nevertheless, the combination is limited in selecting the appropriate UVA/UVB filters to avoid possible negative interactions between the combining agents [25]. Particularly, some organic filters (e.g., PABA, PABA derivatives, and benzophenones) show considerable negative effects, including eczematous dermatitis, burning sensation, and increased risk of skin cancer [24]. Therefore, sunscreens have recently minimized or avoided the use of these compounds to protect consumers from undesirable effects. For example, the use of the two most popular organic filters, octinoxate (ethylhexul methoxycinnamate) and oxybenzone, has recently been restricted in Hawaii because of their negative effect on the coral reefs [27]. Besides, some photo unstable filters (e.g., avobenzone and dibenzoylmethanes) show a number of photoreactive results in the formation of photoproducts that can absorb in different UV regions, therefore reducing their photoprotective efficacy [28–30]. Particularly, these photodegradation products can come in direct contact with the skin, thus promoting phototoxic, photosensitizing, and photoallergic contact dermatitis on the skin [30].

## 2.2. Inorganic UV Filters

Inorganic blockers have been approved to protect human skin from direct contact with sunlight by reflecting or scattering UV radiation over a broad spectra [9]. The current agents are ZnO, $TiO_2$, $Fe_xO_y$, calamine, ichthammol, talc, and red veterinary petrolatum (Figure 3) [11]. Although they are generally less toxic, more stable, and safer for human than those of organic ingredients, they are

visible due to white pigment residues left on the skin and can stain clothes [11]. Since the early 1990s, these metal oxides have been synthesized in the form of micro and nanoscale particles (10–50 nm), which can reduce the reflection of visible light and make them appear transparent throughout the skin, resulting in enhance aesthetics over the larger size [11]. For instance, micro-size $TiO_2$ and ZnO have been replaced nano-size $TiO_2$ and ZnO in sunscreen, eliminating undesired opaqueness and improve SPF value [31].

Moreover, the main disadvantage of utilization of nanoparticles (NPs) is that sunscreens tend to block shorter wavelength from UVAII to UVB rather than long radiation (visible and UVA range). In particular, most NPs can produce ROS radicals and are small enough to penetrate into the stratum corneum, thus causing severe skin effects with prolonged exposure, such as photoallergic contact dermatitis and skin aging [32]. Therefore, in order to improve natural appearance as well as reduce side effects on the skin, these cosmetics using nanoparticles need to be controlled by numerous factors, including particles size and distribution, agglomeration and aggregation, and morphology and structure of the NPs [28]. For examples, the utilization of $TiO_2$ and ZnO NP-coated silicon or doped elements ($Al_2O_3$ and Zr) can minimize ROS production and prevent negative effects as mentioned above [11,33].

### 2.3. Hybrid UV Filters (Organic/Inorganic Agents)

According to the literature, hybrid materials are two half-blended materials intended to create desirable functionalities and properties [13]. They are constituted of organic components (molecule or organic polymer) mixed with inorganic components (meal oxides, carbonates, phosphates, chalcogenides, and allied derivatives) at the molecular or nanoscale [34]. The combination creates ideal materials with a large spectrum and high chemical, electrochemical, optical transparency, magnetic, and electronic properties [13]. Furthermore, some less toxic and biocompatible hybrid materials have been utilized as active ingredients in cosmetics due to their ability to absorb or deliver organic substances into the hair cuticle and skin layers, thereby improving skin care effect [34]. For instance, L'Oréal and Kerastase have introduced the Intra-Cylane$^{TM}$ shampoo, which contains amino functionalized organosilanes hybrid substances that not only protect against hair damage, but also create hair volume expansion, better mechanical properties, and better texture [34]. Merch KgaA and EMD Chemicals Inc. have utilized a number of hybrid compounds, such as silica microcapsules, to control the release of active ingredients that can reduce skin aging and provide high SPF [34].

### 2.4. Botanical Agents

Botanical agents are secondary metabolites produced by living organisms which play a crucial role in the growth and continuity of these organisms [35]. It has been indicated that metabolites possess antioxidant and UV ray absorption abilities [35]. Their featured properties are related to $\pi$-electron systems, which are mainly found in conjugated bond structures expressed in linear chain molecules and in most of aromatic compounds containing electron resonances [35]. Certainly, there is no denying that UV radiation can generate huge amounts of ROS radicals, which leads to inflammation, neutrophil infiltrate activates nicotinamide adenine dinucleotide phosphate (NADPH) oxidase, and sebaceous gland dysfunction, and accelerates skin pigmentation and dermal matrix [36,37]. In the presence of antioxidants, the ROS radicals are directly scavenged and prevented from their biological targets. As a consequence, the propagation of oxidants is limited, resulting in preventing aging [38].

These antioxidant compounds are obtained from vitamin C, vitamin E, and plant extracts (phenolic, carotenoids, and flavonoid compounds) (Figure 3 and Table 1) [39]. In fact, a large number of botanicals have been approved as inactive agents for preserving, emulsifying, moisturizing, and smoothing sunscreen to further protect the skin. These typical varieties are *Aloe vera*, tomatoes, pomegranate, green tea, cucumber, *Pongamia pinnata* (L.)-Indian beech tree, *Spathodea campanulata* (L.)-African tulip tree, *Dendropanax morbifera*, and *Opuntia humifusa* [39–47].

**Table 1.** Photoprotection mechanism of antioxidant compounds.

| Compounds | Protection Mechanism |
| --- | --- |
| Vitamin C | – Neutralizing ROS radicals in aqueous compartments of the skin based on the oxidation capacity of ascorbate [37,48]<br>– Reducing sunburn cell formation, erythema, and immunosuppression [48]<br>– Inhibiting tyrosinase synthesis and maintaining hydration to protect the skin epidermis barrier [37,48]<br>– Challenging: poor skin penetration and instability [48] |
| Vitamin E | – Protecting the cell membrane from oxidative stress [48]<br>– Inhibiting UV-induced cellular damage: photoaging, lipid peroxidation, immunosuppression, and photocarcinogenesis [48,49] |
| Phenolic compounds | – Scavenging free radicals [48]<br>– Conserving proper skin structure through the regulation of matrix metalloproteinases (MMPs) [50]<br>– Inhibiting collagenase and elastase thus facilitating the maintenance of proper skin structure [51]. |
| Flavonoid compounds | – Their double bonds in flavonoid molecules provide a high ability to absorb UV [51]<br>– The presence of hydroxyl groups attached to aromatic rings also contributes to their ROS scavenging capacity [51] |
| Carotenoids | – Physical quenching function: efficacy antioxidants for scavenging peroxide and singlet molecular oxygen ($^1O_2$) radicals generated in during photooxidation [52]<br>– Absorbing of UV, visible, and blue light [52] |

## 2.5. Safety and Health Hazards of Sunscreen Agents

According to the literature, sunscreen agents should be safe, nontoxic, chemically inert, non-irritating, and fully protect against broad spectrum that can prevent photocarcinogenesis and photoaging [53]. However, they also have negative effects, including contact sensitivity, estrogenicity, photoallergic dermatitis, and risk of vitamin D deficiency [54,55].

It has been reported that an increased incidence of melanoma may result from the use of sunscreen. Gorham et al. (2007) pointed out that some commercial sunscreens completely absorb UVB, but transmit large amounts of UVA, which may contribute to risk of melanoma in populations at latitudes greater than 40 °C [56]. In addition, intentional long-term topical sunscreen can increase melanoma risk, especially when using high-SPF products. Thus, the labeling of sunscreen should inform consumers about the carcinogenic hazards related to sunscreen abuse [57].

Moreover, some sensitive ingredients in sunscreen may also be a photoallergic factor. In particular, PABA and oxybenzone are the most common ingredients causing skin disorders [9]. The penetration and systemic toxic effects of inorganic agents at micro- or nano-size have been reported through several in vivo and in vitro analyses. Pan et al. (2009) demonstrated that $TiO_2$ NPs (15 ± 3.5 nm) can pass through cell membranes and impair the function of human dermal fibroblast cultures [58]. Filipe et al. (2006) suggested that $TiO_2$ NPs (~20 nm) in sunscreen appear on the skin surface and in the stratum corneum regions. Therefore, it does not penetrate deeply into the skin [59].

On the other hand, although UVB can cause sunburn for long-term exposure, it is responsible for more than 90% of individual vitamin D production on skin [60]. There are controversies about vitamin

D deficiency due to sunscreen application. In particular, this photoprotection can lead to a significant reduction in the amount of pre-vitamin D3 produced by sunlight in the skin, resulting in insufficient vitamin D levels [60]. In contrast, Fourschou et al. (2012) indicated that vitamin D synthesis increases exponentially with the application of thinner layers of sunscreen (<2 mg/cm$^2$) [55]. On the other hand, Marks et al. (1995) reported adequate production of vitamin D in the Australian population during the summer in most people using sunscreen or without these skin protection substances [61].

## 3. Sunscreen Formulations

### 3.1. Emulsion Sunscreen

An emulsion is termed a lotion or cream depending on its viscosity, respectively, below 50,000 and in the range of 150,000–500,000 centipoises, providing almost unlimited versatility [62]. It is normally produced from two unmixable liquid phases (oil and water), namely "water-in-oil (W/O)" and "oil-in-water (O/W)" emulsions [63]. Moreover, multiple emulsions (O/W/O and W/O/W), containing both O/W and W/O phases in a stable system, show an effective application in recent sun protection technology [64]. Therein, water accounts for the largest proportion, while active ingredients contribute a little amount in an emulsion product. Thus, emulsion sunscreens are cost-effective vehicles [62]. These formulations possess the ability to spread more easily on the skin and disperse from bottles [63]. Further, this formula shows great effectiveness in strategies to achieve high SPF, create a uniform, thick and nontransparent sunscreen film when applied on the skin, and minimize undesirable interaction among active sunscreen ingredients [62,63]. In other words, emulsion sunscreens also provide an elegant medium that can give the skin a smooth and silky feeling without greasy shine [63]. However, these are extremely difficult to stabilize, especially at high temperatures [62].

### 3.2. Gel Sunscreen

Sunscreen gel seems to represent an ideal vehicle from an aesthetic perspective due to its purity and elegance. It is categorized into four main forms, namely aqueous, hydroalcoholic, microemulsion, and oil anhydrous formulations [62].

The aqueous gel must be composed of water and solubilizers (e.g., nonionic surfactants, organic agents, and phosphate esters) at sufficient proportions to ensure the gel will be transparent at all temperatures. Therefore, it is easily washed away when exposed to water or sweat [62,65]. Although organic active molecules (e.g., octyldimethyl PABA or octyl *p*-methoxycinnamate) are primarily attributed to the formula, they are used in low doses due to their high levels of carcinogenicity [65]. Interestingly, the high concentration of organic filters is primarily responsible for increasing the SPF value. Thus, the aqueous gel provides low SPF compared to other kinds of gel sunscreens [65].

The hydroalcoholic gels are formulated by alcohol (ethanol) in conjunction with water, which are important in reducing additional solutes because most lipophilic ingredients are readily miscible in alcohol [66]. This form can provide the desired cooling effect, which is especially refreshing when applied to the skin on summer days [62]. However, this formulation also shows some negative aspects, such as quickly being washed way in the water, causing facial or eye sting on certain individuals, and providing low SPF [62,66].

The microemulsion gels are composed of small particles, allowing them to appear smooth, thick, and evenly on the skin, thus delivering an elegant feel and high SPF [62,67]. Unfortunately, it is markedly expensive to achieve transparent microemulsions containing high-level emulsifiers (15–25%) [62]. Particularly, most emulsifiers are irritating components, so this emulsion system pose a risk to human health [62]. In addition, high emulsification proportion results in reduced water-resistance of these sunscreen products.

The oil anhydrous formula possesses many attributes similar to ointments. However, oil anhydrous products are clear, while the ointments are translucent [62,66]. These products can be produced as a

gel by combining mineral oil and special silica [62,68]. However, they are not widely sold because they are difficult to produce and quite expensive.

### 3.3. Aerosol Sunscreen

In addition to lotions and creams, aerosol sunscreens are topically applied to protect skin disorders from harmful sunlight. These products can be easily spread onto the surface of skin, and distribute active ingredients to form a thin film on the skin [69]. However, this application may result in the uneven spreading of sunscreen agents, corresponding to some high-coverage areas with an excessive amount of sunscreen and other areas with little coverage to protect the skin satisfactorily [69]. Nevertheless, the aerosol products have not become as popular as other sunscreens due to some critical negative aspects. First, they are typically oil-based, making them quite expensive and often reducing their effectiveness [62]. In addition, it is hard observe where the sunscreen has been applied. Caution must be taken to avoid accidentally spraying sunscreen into the eyes [62].

### 3.4. Sun Stick

The sun stick is undoubtedly one of the most convenient products due to its small size and light weight. The sun stick is produced by two main emulsion components, namely oil and oil-soluble components, through the incorporation of petrolatum and waxes. Thus, it tends to have a greasy feel on the skin, which is a common problem of most water-resistant sunscreens [62]. However, this product has gained great attention due to its ability to cover a very small surface area during each application. It is also easy to carry and re-touch [14]. This form is subdivided into three categories, namely transparent, semi-transparent, and matte sunscreen [14]. The transparent formula contains only chemical UV filters, while semi-transparent is formulated mainly by chemical and mineral substances and matte is composed of only mineral sunscreen ingredients [14].

## 4. Novel Properties of Commercial Sun Protection Products

### 4.1. Sunscreen with Antioxidants and Anti-Aging

Regarding the beneficial effects of natural agents, many sunscreens have been produced by combining one or more natural ingredients (e.g., extracts and nutrient compositions) and conventional ingredients (e.g., $TiO_2$, ZnO, and benzoate derivatives) [70]. Particularly, these products have been found to be safe and are able to overcome undesirable effects by reducing the utilization of inorganic and organic compounds [70]. For instance, US patent No. 8,337,820B2 disclosed a water-soluble sunscreen formula, mainly including $TiO_2$ and 5-hydroxy-trytophan extracted from *Griffonia simplicifolia*, which can protect individuals with rosacea or other sensitive skin types from harmful UV radiation. This formulation does cause skin disturbances, such as inflammation, erythema, and flushing, because it does not contain organic ingredients [71]. In another study, US patent No. 6,440,402B1 revealed a synergistic absorption effect of *Kaempferia galangal* (ginger) root extract upon prolonged exposure to sunlight. It has been suggested that the topical sunscreen comprising an active agent, a cosmetically vehicle, and sufficient amount of *K. galangal* can enhance photostability and sunscreen efficacy [34]. In fact, the Tomato Lycopene (SPF 20) sunscreen from 100% Pure contains a large amount of lycopene that can protect skin from pollution effects (wrinkles and aging) and provide a moisturizing feeling. The sunscreen of Blossom Kochhar Aroma Magic has been innovated by taking advantage of the cucumber's preeminent features, like the variety of vitamins, and non-greasy and skin-friendly features, to improve protection effects and prevent visible signs of aging on the skin.

### 4.2. Sunscreen Combined with DNA Repair Enzymes

There is no denying the fact that UV radiation can penetrate deep into the skin and damage cells where skin cancer originates [72]. The consequences are considerably enormous while damaged DNA is unrepaired, especially after decades of repeated damage, including tone loss, hyperpigmentation,

wrinkle formation, and skin cancer [73]. In the early stages, damage appears as various symptoms, including texture and tone loss, hyperpigmentation, and wrinkle formation. In the end stages, skin cancers may result [12,73]. On the other hand, traditional sunscreens only represent a "passive photoprotection" and are not effective after damaging skin cells due to sun exposure [73]. Therefore, an "active photoprotection" approach has been invented by combining antioxidants and liposome-containing DNA repair enzymes, which could be an advanced photo-strategy to fill the current gap in sun protection [73].

The topical application of DNA repair enzymes serves to complement the internal DNA repair mechanisms [73]. The direct repair of damaged DNA using endogenous repair enzymes can reduce the mutation rates and strengthen the immune response to tumor cells [73]. In order to prevent the incidence, as well as reduce healthcare expenses in skin treatment, a number of products have recently focused on not only blocking UV, but also boosting DNA repair and modulating DNA transcription [72]. The breakthrough DNA repair technology was developed based on chemopreventive benefits of topical T4 endonuclease (T4N5), photolyase, combining photolyase with endonuclease, and 8-oxoguanine glycosylase [72] (Table 2). In fact, Emanuele et al. (2013) successful applied photolyase from *Aspergillus nidulans* and endonuclease from *Micrococcus luteus* as xenogenic DNA repair enzymes to reverse the molecular events related to skin aging and carcinogenesis due to UV radiation exposure [74]. These enzymes were used to abrogate telomere shortening and c-FOS gene hyperexpression on the skin, consequently preventing skin aging [74]. In another study, Carducci et al. (2015) compared the protective effects of traditional sunscreen alone and those of plus DNA repair enzymes in 28 patients with actinic keratosis [75]. Interestingly, the results indicated that the sunscreen in combination with DNA repair enzymes may outperform those of traditional in reducing cancerization and UV-related molecular signatures in the patients. It was suggested that DNA repair enzymes are more effective in preventing malignant transformation into invasive against squamous cell carcinoma [75].

**Table 2.** DNA repair enzymes for skin protection application.

| DNA Repair Enzymes | Proposal Mechanism and Proven Effects |
|---|---|
| Topical T4 endonuclease | – Enhancing DNA repair by eliminating cyclobutane pyrimidine dimers (CPDs) [76] <br> – Reducing of precancerous and cancerous in high-risk individuals [77] |
| Photolyase | – Using energy from blue light to quickly repair damaged DNA by catalyzing electron transfer reactions, resulting in the splitting of cyclobutane rings [78,79] <br> – Reducing UV-induced CPDs and precancerous lesions in humans [80] |
| 8-Oxoguanine glycosylase | – Identifying and initiating repairing DNA photo-lesion (8-oxo-7,8-dihydroguanine) caused by ROS [81] <br> – Reducing of UVB-induced tumor progression in mice [82] |

### 4.3. Sunscreen Against Environmental Pollutants

In addition to UV rays, pollutants, such as particulate matter (PM), polyaromatic hydrocarbons, sulfur oxides ($SO_2$), and nitrogen oxides ($NO_x$), can negatively affect skin. It has been indicated that these pollutants can induce inflammation, hyperpigmentation, and collagen breakdown, leading to skin dryness, dark spots, loss of firmness, uneven skin tone, aggravation of acne, and wrinkle formation [13]. Therefore, manufacturers have countered this issue by supplementing antioxidant ingredients to cosmetic products that can minimize and prevent side effects. The protecting mechanism reduces inflammation, prevents particle load on skin by cleansing or exfoliation, and promotes

collagen/elastin synthesis [13]. A few examples of products which deliver anti-pollution benefits by the above mechanisms are "Dr Dennis Cross Dark Sot Sun Defense Broad Spectrum SPF 50" with melatonin defense complex and "Clarins UV plus anti-pollution SPF 50 Broad Spectrum" sunscreen with white tea extract [13]. The Clarins's product is believed to help fight pollutants due to its lightweight, oil-free cream, which combines SPF with extracts from organic cantaloupe and the alpine plant.

### 4.4. Sunscreen Against Blue Light

Blue light (380–500 nm) is derived from sunlight or electronic devices such as smartphones, tablets, and computers [83]. Due to its high energy, it is useful on photodynamic therapy when using a combination of photosynthesizing drug and a high-intensity light source to treat cancer [84,85]. However, when blue light enters deep into the skin, it can cause deleterious effects all skin layers by generating ROS and weakening the epidermal barrier, thereby damaging the extracellular matrix and accelerating aging [14,15]. Therefore, it is necessary to protect skin against blue light. Recently, some sunscreen products have improved their ability to protect against blue light, such as "Sun Expertise (SPF 50+)" and "City Skin Age Defense (SPF 50 and PA+++)" from SKEYDOR and Murad, respectively. It has been suggested that UV filters can break through the boundaries of UVB and UVA to continue into the blue light spectrum. UV filters have also been suggested to contain vitamins and microalgae that can enhance the skin's defense [14].

### 4.5. Sunscreen against Thermal IR

IR rays, accounting 54.3% of total solar radiation reaching the Earth, have also been proposed to be deleterious to human skin [16,86]. Regarding its ability to penetrate the epidermis, dermis, and subcutaneous tissues, this radiation can be a detrimental factor that damages collagen content of the skin through the creation of ROS radicals and increased MMP-1 and MMP-9 activity in the same manner of UV rays [16,86]. Therefore, the application of appropriate antioxidants has been considered as an effective photoprotective strategy against these impacts. In fact, the IR protection effectiveness of sunscreen has been investigated through the in vivo method. Kim et al. (2019) evaluated the resistance to IR rays provided by sunscreen in 155 Korean volunteers, and recorded the IR reflectance of volunteer's skin using a benchtop model of an IR light source and a reflectance measuring probe [87]. The results showed that the infrared protection factor (IPF) in protected skin was greater than that of unprotected skin. This study also demonstrated good correlation between IPFs and inorganic sunscreen ingredients in commercial cosmetics [87]. Recently, a number of defense-IR-protection sunscreens have been launched that offer skin protection from sun damage and also prevent aging due to chronic exposure to IR. For example, the "Total defense and repair SPF 34" from Skin Medica is produced using an advanced antioxidant complex. This restorative formula goes beyond UVA and UVB protection, reducing the appearance of fine lines and wrinkles.

## 5. Conclusions and Outlook

The use of sunscreen is beneficial in minimizing skin disorders caused by UV radiation and other factors, especially in people with fair skin. Young adults are regularly advised to use sunscreen to avoid or minimize photodamaging effects. The FDA also suggests that consumers should reapply sunscreen (2 mg/cm$^2$) at least every 2 h, or more often if consumers are sweating or jumping in and out of the water [88]. Many recently produced commercial products contain mot only conventional active ingredients (TiO$_2$, ZnO, PABA, and salicylates), but also inactive components (botanicals, vitamins, and hybrid materials). These antioxidants can potentially eliminate the release of oxidants induced by UV rays and the metallic oxide components in commercial products. In addition, these products have been developed using a variety of comprehensive formulations (e.g., emulsion, gel, aerosol, and stick) to provide long-lasting protection, spreading, moisturizing, and high stability. In addition, these novel products are often produced by a high number of UV filters combined with botanicals, vitamins, DNA repair enzymes, and film-forming polymers that can contribute to a higher SPF. Particularly, recent

sunscreens also possess the outstanding capacity to address the demands of consumers to protect the skin from all environmental aggressors, such as UV rays, pollution, blue light, and IR.

Along with the above-mentioned novel features, sunscreen is currently required to produce non-sticky or lighter textures that can provide longer-lasting protection, making sunscreen more convenient in daily routines, especially sport and water activities, and even in warm and humid climates. Therefore, greater innovations in discovering and synthesizing novel components (e.g., polymers, nanomaterials, and botanicals) and higher demand for lighter emollients are expected to continue in the future. Besides, manufacturers must overcome current critical challenges, such as the increasing incidence of melanoma risk, and eczema or photo allergies due to sensitization reactions between chemicals and the skin, in order to optimize sun protection effect.

**Author Contributions:** Y.-C.L. planned the study and contributed the main ideas; L.T.N.N. and Y.-C.L. were principally responsible for the writing of the manuscript; Y.-C.L., V.V.T., D.P., J.-Y.M., and M.C. commented on and revised the manuscript.

**Funding:** This work was supported by Basic Science Research Program through the National Research Foundation of Korea funded by the Ministry of Education (NRF-2017R1D1A1A09000642), by a grant from R & D program of the Korea Railroad Research Institute (KRRI), Korea, and also by Biocell Korea Co., Ltd.

**Conflicts of Interest:** The authors declare no conflict of interest.

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
