# Peer review of "Recent Trends of Sunscreen Cosmetic: An Update Review"

_cosmetics, doi:10.3390/cosmetics6040064_

Round 1
Reviewer 1 Report
This manuscript is a review written by non-experts of UV and sunscreen, and reject is appropriate.
The skin penetration of UVB in Figure 1 is incorrect.
Line 73-79; The description is significantly wrong. Authors should refer to recent articles.
Figure 2 is wrong.
Author Response
Cosmetics
Manuscript ID: cosmetics-619444
Title: Recent Trends of Sunscreen Cosmetic: An Update Review
Authors: Le Thi Nhu Ngoc, Vinh Van Tran, Ju-Young Moon, Minhe Chae, Duckshin Park*, Young-Chul Lee*
Reviewer 1:
This manuscript is a review written by non-experts of UV and sunscreen, and reject is appropriate.
The skin penetration of UVB in Figure 1 is incorrect.Response: Thank you for your comment. The penetration of UVB into skin layers has been corrected.
Figure 1. Electromagnetic spectra solar UV rays and their biologic effects on the skin [2]
Line 73-79: The description is significantly wrong. Authors should refer to recent articles.Response: Thank you for your kind comments. However, it is believed that the description is really correct because the information of Protection Grade of UVA (PA) has been carefully referred from published papers [1,2]. In the revised manuscript, the paragraph has been rewritten in order to improve the clarity of this description.
“Furthermore, the photoprotective efficiency of sunscreen is determined through sun protection factor (SPF) and the protection grade of UVA (PA) values. According to Food and Drug Administration (FDA) regulations, commercial products must be labeled with SPF values that indicate how long they will protect the user from UV radiation as well as show the effectiveness of protection [17]. Certainly, the SPF values are generally in the range of 6-10, 15-25, 30-50, and 50+, corresponding to low, medium, high, and very high protection, respectively [17]. Nevertheless, there are some fundamental misunderstandings of the SPF. Some argument is that an SPF 15 sunscreen can absorb 93% of the erythemogenic UV radiations, while an SPF 30 product can block 96%, just over 3% (Figure 2) [18]. The argument may be correct when evaluating sun protection capacity, but this is not sufficient in assessing the amount of UV radiation entering the skin. In other word, half as much will penetrate into the skin when applied SPF 30, compared to SPF 15 product, a different factor 2 [18]. This is also illustrated by comparing an SPF 10 with an SPF 50 sunscreen. Ten and two photons transmit (%) through sunscreen film and then enter the skin applied SPF 10 and 50 products, respectively, a difference of 5 as it is expected [18]. On the other hand, the Japan Cosmetic Industry Association (JCIA) has developed an in-vivo persistent pigment darkening (PPD) method to evaluate UVA efficacy of sunscreen since 1996 [19]. Corresponding to the level of Protection Grade of UVA (PA) obtained from the PPD test, sunscreens will be labeled with PA+, PA++, PA+++, and PA++++ [19,20]. In which, the labeled PA+ expresses low protection, mainly contributed by two or more but less than 4 UVA filters. Sunscreens containing four to eight sunscreen agents show moderate levels of UVA blocking and are labeled as PA++. In contrast, the PA+++ and PA++++ symbols represent products that are composed of more than eight UVA filters and provide a high sunscreen efficacy [10,19,20].
Figure 2. Illustration of the SPF definitions, including filtered and transmitted UV radiation [18]”
Figure 2 is wrong.Response: Thank you for your comment. However, the figure has been reprinted as well as upgraded from published papers (Figure S1 and S2), therefore it can be believed that the figure actually provides accurate information on the classification of sunscreen agents [3,4]. Please look at the references below to confirm this accuracy.
Figure S1. Classification of sunscreen agents [3]
Figure S2. Classification of topical and systemic photoprotective agents [4]
Besides, in the revised manuscript, Figure 2 has been corrected and upgraded, according to other reviewer’s suggestions.
“Figure 3. Sunscreen agent classification [23]”
References
[1] Moyal, D. (2010). UVA protection labeling and in vitro testing methods. Photochem. Photobiol. Sci. 9(4), 516-523.
[2] Donglikar, M. M., & Deore, S. L. (2016). Sunscreens: A review. Phcog. J. 8(3).
[3] Latha et al. (2013) Sunscreening agents: a review. J. Clin. Aesthet. Dermatol. 6(1),16.
[4] Kaur et al. (2014) Need of UV protection and evaluation of efficacy of sunscreens. J. Cosmet. Sci. 65(5), 315-345.

Reviewer 2 Report
This is a good review of the current status of sunscreens. The authors have included many interesting and timely topics in the review, including: 1) the need for protection from blue light and IR, 2) the use and potential problems of nanoparticles in sunscreens, 3) the value of antioxidants in sun protection products to reduce ROS and inflammation, and 4) the potential risk of developing melanoma from using just a UVB sunscreen.
The manuscript was easy to read, but does need some English editing.
The authors mention the use in Japan of a PA rating system, but I could not find what the letters PA mean. the authors might want to mention what this stands for.
I found no substantive issues with the manuscript.
Author Response
Cosmetics
Manuscript ID: cosmetics-619444
Title: Recent Trends of Sunscreen Cosmetic: An Update Review
Authors: Le Thi Nhu Ngoc, Vinh Van Tran, Ju-Young Moon, Minhe Chae, Duckshin Park*, Young-Chul Lee*
Reviewer 2:
This is a good review of the current status of sunscreens. The authors have included many interesting and timely topics in the review, including: 1) the need for protection from blue light and IR, 2) the use and potential problems of nanoparticles in sunscreens, 3) the value of antioxidants in sun protection products to reduce ROS and inflammation, and 4) the potential risk of developing melanoma from using just a UVB sunscreen.
The manuscript was easy to read, but does need some English editing.
The authors mention the use in Japan of a PA rating system, but I could not find what the letters PA mean. The authors might want to mention what this stands for.
I found no substantive issues with the manuscript.
Response: Thank you so much for your kind comments. Actually, the entire manuscript has been seriously edited by a native speaker. Please look at the English edition certificate below.
Besides, the PA letter is a stand for Protection Grade of UVA. The PA rating system ranges from “PA+” (least) to “PA++++” (most). Although there is no UVA protection rating system currently recognized by the Food and Drug Administration (FDA), the PA system has been approved in some Asian countries, especially in Japan. According to improve the clarity of the sun protection factor (SPF) and PA system, this paragraph has been rephrased and inserted essential information.
“Furthermore, the photoprotective efficiency of sunscreen is determined through sun protection factor (SPF) and the protection grade of UVA (PA) values. According to Food and Drug Administration (FDA) regulations, commercial products must be labeled with SPF values that indicate how long they will protect the user from UV radiation as well as show the effectiveness of protection [17]. Certainly, the SPF values are generally in the range of 6-10, 15-25, 30-50, and 50+, corresponding to low, medium, high, and very high protection, respectively [17]. Nevertheless, there are some fundamental misunderstandings of the SPF. Some argument is that an SPF 15 sunscreen can absorb 93% of the erythemogenic UV radiations, while an SPF 30 product can block 96%, just over 3% (Figure 2) [18]. The argument may be correct when evaluating sun protection capacity, but this is not sufficient in assessing the amount of UV radiation entering the skin. In other word, half as much will penetrate into the skin when applied SPF 30, compared to SPF 15 product, a different factor 2 [18]. This is also illustrated by comparing an SPF 10 with an SPF 50 sunscreen. Ten and two photons transmit (%) through sunscreen film and then enter the skin applied SPF 10 and 50 products, respectively, a difference of 5 as it is expected [18]. On the other hand, the Japan Cosmetic Industry Association (JCIA) has developed an in-vivo persistent pigment darkening (PPD) method to evaluate UVA efficacy of sunscreen since 1996 [19]. Corresponding to the level of Protection Grade of UVA (PA) obtained from the PPD test, sunscreens will be labeled with PA+, PA++, PA+++, and PA++++ [19,20]. In which, the labeled PA+ expresses low protection, mainly contributed by two or more but less than 4 UVA filters. Sunscreens containing four to eight sunscreen agents show moderate levels of UVA blocking and are labeled as PA++. In contrast, the PA+++ and PA++++ symbols represent products that are composed of more than eight UVA filters and provide a high sunscreen efficacy [10,19,20].
Figure 2. Illustration of the SPF definitions, including filtered and transmitted UV radiation [18]”

Reviewer 3 Report
The focus of this review is of great interest. However, extensive editing of English language and style is needed to improve the readability of the manuscript, that in my opinion is one of the most important feature a review should have. So authors should carefully re-read (and possibly re-write) the entire manuscript before it can be accepted for publication.
The following are some other suggestions for further improving, in my opinion, the quality of the paper:
Figure 1: I would suggest authors to attribute each biological effect to the proper UV radiation. Page 3, lines 71-79: more details should be added to make this paragraph clearer. Figure 2: what does “Ce-aminoclay” stand for? And why is a distinction made between “topical” and “botanicals”? Are not botanicals applied topically? Please clarify. Moreover, what is the role of antimalarials in sunscreen agents? Page 4, lines 103-108 and 128-130: these sentences (among others) are very hard to understand, please rephrase them. Table 1: I think that also phenolic compounds should be considered as ROS scavengers. Lines 177-179: is the mechanism by which sunscreens can increase melanoma risk known? Page 7, lines 223-225 and 248-250: please rephrase the sentences, they are not clear.
Author Response
Cosmetics
Manuscript ID: cosmetics-619444
Title: Recent Trends of Sunscreen Cosmetic: An Update Review
Authors: Le Thi Nhu Ngoc, Vinh Van Tran, Ju-Young Moon, Minhe Chae, Duckshin Park*, Young-Chul Lee*
Reviewer 3:
The focus of this review is of great interest. However, extensive editing of English language and style is needed to improve the readability of the manuscript that in my opinion is one of the most important features a review should have. So authors should carefully re-read (and possibly re-write) the entire manuscript before it can be accepted for publication.Response: Thank you for your kind comments. Actually, the manuscript has been upgraded and checked on English by a native speaker. Please look at the English edition certificate below.
The following are some other suggestions for further improving, in my opinion, the quality of the paper:2.1. Figure 1: I would suggest authors to attribute each biological effect to the proper UV radiation.
Responses: Thank you for your kind comments. According to your recommendation, the biological effects have been attributed, according to the consequences to exposure of each proper UV radiations. Besides, these effects have been provided in Introduction section
“Recently, UV-induced skin damage is one of the most common concerns in the world. Certainly, UVA is a risk of skin aging, dryness, dermatological photosensitivity, and skin cancer. It damages DNA through the generation of reactive oxygen species (ROS), which causes oxidative DNA base modifications and DNA strand breaks, resulting in mutation formation in mammalian cells [1,2,8]. On the other hand, UVB can directly damage DNA through the formation of pyrimidine dimer and then cause apoptosis or DNA replication errors, leading to mutation and cancer [5]. Besides, although UVC is the shortest and most energetic wavelength, it is the most dangerous UV rays because it can cause various adverse effects (e.g., mutagenic and carcinogenic) [5]. However, UVC does not penetrate through the atmosphere layer.”
Figure 1. Electromagnetic spectra solar UV rays and their biologic effects on the skin [2]
2.2. Page 3, lines 71-79: more details should be added to make this paragraph clearer.
Response: Thank you for your comment. This paragraph has been upgraded in order to improve its rationality and accuracy.
“Furthermore, the photoprotective efficiency of sunscreen is determined through sun protection factor (SPF) and the protection grade of UVA (PA) values. According to Food and Drug Administration (FDA) regulations, commercial products must be labeled with SPF values that indicate how long they will protect the user from UV radiation as well as show the effectiveness of protection [17]. Certainly, the SPF values are generally in the range of 6-10, 15-25, 30-50, and 50+, corresponding to low, medium, high, and very high protection, respectively [17]. Nevertheless, there are some fundamental misunderstandings of the SPF. Some argument is that an SPF 15 sunscreen can absorb 93% of the erythemogenic UV radiations, while an SPF 30 product can block 96%, just over 3% (Figure 2) [18]. The argument may be correct when evaluating sun protection capacity, but this is not sufficient in assessing the amount of UV radiation entering the skin. In other word, half as much will penetrate into the skin when applied SPF 30, compared to SPF 15 product, a different factor 2 [18]. This is also illustrated by comparing an SPF 10 with an SPF 50 sunscreen. Ten and two photons transmit (%) through sunscreen film and then enter the skin applied SPF 10 and 50 products, respectively, a difference of 5 as it is expected [18]. On the other hand, the Japan Cosmetic Industry Association (JCIA) has developed an in-vivo persistent pigment darkening (PPD) method to evaluate UVA efficacy of sunscreen since 1996 [19]. Corresponding to the level of Protection Grade of UVA (PA) obtained from the PPD test, sunscreens will be labeled with PA+, PA++, PA+++, and PA++++ [19,20]. In which, the labeled PA+ expresses low protection, mainly contributed by two or more but less than 4 UVA filters. Sunscreens containing four to eight sunscreen agents show moderate levels of UVA blocking and are labeled as PA++. In contrast, the PA+++ and PA++++ symbols represent products that are composed of more than eight UVA filters and provide a high sunscreen efficacy [10,19,20].
Figure 2. Illustration of the SPF definitions, including filtered and transmitted UV radiation [18]”
2.3. Figure 2: what does “Ce-aminoclay” stand for? And why is a distinction made between “topical” and “botanicals”? Are not botanicals applied topically? Please clarify. Moreover, what is the role of antimalarials in sunscreen agents?
Response: Thank you for your kind comments. Actually, the phrase “Ce-aminoclay” presented in the hybrid sunscreen agent represents a novel hybrid material for future sunscreen application. However, the phrase “Ce-aminoclay” was removed in the figure after careful consideration.
In fact, there are two approaches (topical and systemic) to protect the skin from the harmful effects of solar radiation. In which, topical sunscreen ingredients, as known as active agents, include chemical agents in the form of solutions, gels, cream, or ointments that reduce the deleterious effects of the skin to excessive exposure to UV rays. They protect the skin by absorbing, reflecting, and scattering UV radiations. Meanwhile, sun-protective systemic preparations (i.e., inactive ingredients), such as botanical agents and vitamins, which can act as quenchers of singlet oxygen and/or free radicals, or as stabilizers of the cell membrane, as well as repair DNA damage due to UV exposure. The phrase “Botanical” presented in this figure is a kind of systemic agents, and it has been replaced by the term “systemic” after careful consideration (Figure 3).
“Figure 3. Sunscreen agent classification [23]”
Antimalarials (chloroquine) are valuable therapeutic agents but are not first-line drugs of choice and should be used only after conservative measures (topical sunscreen) have been tried [1]. These include lupus erythematosus (LE), polymorphous light eruptions (PMLE), solar urticaria, and porphyria cutanea tarda (PCT). It has been demonstrated that chloroquine in low doses can work merely as sunscreens that filter radiation and anti-inflammatory. Besides, treatment with antimalarial drugs, chloroquine, and hydroxychloroquine is currently being advocated as an alternative to phlebotomy in selected patients to eliminate the biochemistry of the disease and its associated symptoms on the skin [1].
2.4. Page 4, lines 103-108 and 128-130: these sentences (among others) are very hard to understand, please rephrase them.
Response: Thank you for your kind comments. These sentences have been rewritten to improve the clarity.
“2.1. Organic UV filters
Organic blockers are classified into either UVA (anthranilates, dibenzoylmethanes, and benzophenones) or UVB filters (salicylates, cinnamates, PABA derivatives, and camphor derivatives) which play an important role in absorption activity of sunscreen (Figure 3) [24]. These agents show outstanding aesthetic properties, including stability, nonirritant, nonvolatile, non-photosensitizing, and non-staining to human skin, compared to inorganic UV filters [25]. Besides, they are mostly used in combination at levels currently allowed by the FDA to provide broad-spectrum absorption as well as increased SPF values [26]. Nevertheless, the combination is limited in selecting the appropriate UVA/UVB filters to avoid possible negative interactions between the combining agents [25]. Particularly, some organic filters (e.g., PABA, PABA derivatives, and benzophenones) show considerable negative effects, including causing eczematous dermatitis, burning sensation, and increased risk of skin cancer [24]. Therefore, sunscreens have recently minimized or avoided the use of these compounds to protect consumers from undesirable effects.”
“Therefore, in order to improve natural appearance as well as reduce side effects on the skin, these NPs using cosmetics need to be controlled by the following factors, including particles size and distribution; agglomeration and aggregation; morphology and structure of the NPs [28]. For examples, the utilization of TiO2 and ZnO NPs coated silicon or doped other elements (Al2O3 and Zr) can minimize ROS production and avoid negative effects as mentioned above [11,27].”
2.5. Table 1: I think that also phenolic compounds should be considered as ROS scavengers.
Response: Thank you for your suggestion. The scavenging ROS radical activity of phenolic compounds has been inserted (Table 1).
Table 1. Photoprotection mechanism of anti-oxidant compounds
|
Compounds |
Protection mechanism |
|
Vitamin C |
- Neutralizing ROS radicals in aqueous compartments of the skin based on the oxidation capacity of ascorbate [33,44] - Reducing sunburn cell formation, erythema, and immunosuppression [44] - Inhibiting tyrosinase synthesis and maintaining hydration to protect the skin epidermis barrier [33,44] - Challenging: poor skin penetration and instability [44] |
|
Vitamin E |
- Protecting the cell membrane from oxidative stress [44] - Inhibiting UV-induced cellular damage: photoaging, lipid peroxidation, immunosuppression, and photocarcinogenesis [44,45] |
|
Phenolic compounds |
- Scavenging free radicals [44] - Conserving proper skin structure through the regulation of matrix metalloproteinases (MMPs) [46] - Inhibiting collagenase and elastase thus facilitating the maintenance of proper skin structure [47]. |
|
Flavonoid compounds |
- Their double bonds in flavonoid molecules provide a high ability to absorb UV [47] - The presence of hydroxyl groups attached to aromatic rings also contributes to their ROS scavenging capacity [47] |
|
Carotenoids |
- Physical quenching function: efficacy anti-oxidants for scavenging peroxide and singlet molecular oxygen (1O2) radicals generated in during photooxidation [48] - Absorbing of UV, visible, and blue light [48] |
2.6. Lines 177-179: is the mechanism by which sunscreens can increase melanoma risk known?
Response: Thank you for your reasonable comment. According to relevant studies, it has been shown that sunscreen use by sun-sensitive consumers when intentionally exposed to sun rays can increase the duration of exposure without reducing sunburn occurrence. This increased duration could be the reason why the risk of melanoma increases when applied sunscreen. Hence, advertising and labeling of sunscreen should inform consumers about the cancer risk associated with sunscreen abuse [2].
2.7. Page 7, lines 223-225 and 248-250: please rephrase the sentences, they are not clear
Response: Thank you for your recommendation. In this revised manuscript, the sentence in lines 223-225 has been rewritten to make it clearer, while the sentence (lines 248-250) has been removed because it is unnecessary and irrelevant to the whole paragraph.
“To begin with, the aqueous gel must be composed of water and solubilizers (e.g., nonionic surfactants, organic agents, and phosphate esters) at sufficient proportions to ensure that the gel will be transparent at all temperature; therefore, it is easily washed away when exposed to water or sweat [58,61]. Although organic active molecules (e.g., octyldimethyl PABA or octyl p-methoxycinnamate) are primarily attributed to the formula, they are used in low doses due to their high levels of carcinogenicity [61]. Interestingly, the high concentration of organic filters are major responsible for increasing the SPF value, thus, the aqueous gel just offers low SPF, compared to other kinds of gel sunscreens [61].”
References
[1] Madhu A.P. (1982), Sunscreens: Topical and systemic approaches for protection of human skin against harmful effects of solar radiation. J. Am. Acad. Dermatol. 7(3), 285-312.
[2] Kerr, A.; Ferguson, J. Photoallergic contact dermatitis. Photodermatol. Photoimmunol. Photomed. 2010, 26, 56-65.

Round 2
Reviewer 1 Report
The sentecse in abstract "Particularly, these products have been supplemented with several preeminent properties to protect against the negative effects of not only UV but also infrared (IR) radiation, blue light, and pollution." should be rewrite to the proper sentence "Particularly, these products have been supplemented with several preeminent properties to protect against the negative effects of not only UVB but also UVA.", because all products do not supplement with infrared (IR) radiation, blue light, or pollution.
The following sentense should be add to 2.1. "The use of the two most common organic filters, oxybenzone and octinoxate (ethylhexyl methoxycinnamate), has recently been restricted in Hawaii due to their harmful effect on the coral reefs (proper references)."
Also, it should be added about decrease of the efficacy of Avobenzone etc by the photo degradation and its skin toxicity (proper references).
Author Response
Manuscript ID: cosmetics-619444
Type of manuscript: Review
Title: Recent Trends of Sunscreen Cosmetic: An Update Review
Authors: Le Thi Nhu Ngoc, Vinh Van Tran, Ju-Young Moon, Minhe Chae, Duckshin Park, Young-Chul Lee *
Reviewer 1:
The sentence in abstract "Particularly, these products have been supplemented with several preeminent properties to protect against the negative effects of not only UV but also infrared (IR) radiation, blue light, and pollution." should be rewrite to the proper sentence "Particularly, these products have been supplemented with several preeminent properties to protect against the negative effects of not only UVB but also UVA.", because all products do not supplement with infrared (IR) radiation, blue light, or pollution.Response: Thank you for your suggestion. The sentence in the abstract has been rewritten according to your recommendation.
“Ultraviolet (UV) radiation has been demonstrated to cause skin disorders, including sunburn and relative symptoms of prolonged exposure. It has been reported that sunscreens have beneficial effects in reducing the incidence of skin disorders (sunburn, skin aging, and immunosuppression) through its ability to absorb, reflect, and scatter UV. Many commercial products have recently been manufactured from not only usual organic and inorganic UV filters but also hybrid and botanical ingredients using typical formulations (emulsion, gel, aerosol, and stick). Particularly, these products have been supplemented with several preeminent properties to protect against the negative effects of not only UVB but also UVA. However, the use of sunscreen has faced many challenges, including inducing photoallergic dermatitis, environment pollution, and deficiency of vitamin D production. Therefore, consumers should efficiently apply suitable products to improve sun protection as well as avoid the side effects of sunscreen.”
The following sentence should be added to 2.1. "The use of the two most common organic filters, oxybenzone and octinoxate (ethylhexyl methoxycinnamate), has recently been restricted in Hawaii due to their harmful effect on the coral reefs (proper references)."Also, it should be added about decrease of the efficacy of Avobenzone etc by the photo degradation and its skin toxicity (proper references).
Response: Thank you for your kind comments. Section 2.1 has been upgraded with more detailed information about the disadvantages of some popular organic UV filters.
“Organic blockers are classified into either UVA (anthranilates, dibenzoylmethanes, and benzophenones) or UVB filters (salicylates, cinnamates, PABA derivatives, and camphor derivatives) which play an important role in absorption activity of sunscreen (Figure 3) [24]. These agents show outstanding safety and aesthetic properties, including stability, nonirritant, nonvolatile, non-photosensitizing, and non-staining to human skin, compared to inorganic UV filters [25]. Besides, they are mostly used in combination at levels currently allowed by the FDA to provide broad-spectrum absorption as well as increased SPF values [26]. Nevertheless, the combination is limited in selecting the appropriate UVA/UVB filters to avoid possible negative interactions between the combining agents [25]. Particularly, some organic filters (e.g., PABA, PABA derivatives, and benzophenones) show considerable negative effects, including causing eczematous dermatitis, burning sensation, and increased risk of skin cancer [24]. Therefore, sunscreens have recently minimized or avoided the use of these compounds to protect consumers from undesirable effects. For example, the use of the two most popular organic filters, octinoxate (ethylhexul methoxycinnamate) and oxybenzone, has recently been restricted in Hawaii because of their negative effect on the coral reefs [27]. Besides, some photounstable filters (e.g., avobenzone and dibenzoylmethanes) show a number of photoreactivity resulting in the formation of photoproducts that can absorb in different UV regions, therefore reducing their photoprotective efficacy [28-30]. Particularly, these photodegradation products can come in direct contact with the skin, thus promoting phototoxic, photosensitizing, and photoallergic contact dermatitis on the skin [30].”

Reviewer 3 Report
I appreciated authors’efforts to improve the quality of the manuscript and their answer to my concerns, so I can now recommend publication of the manuscript. I am just still a little concerned about the use of the term “systemic” for botanical agents. Maybe authors should think about the possibility to use a different term, such as “coadjuvant”.
Author Response
Manuscript ID: cosmetics-619444
Type of manuscript: Review
Title: Recent Trends of Sunscreen Cosmetic: An Update Review
Authors: Le Thi Nhu Ngoc, Vinh Van Tran, Ju-Young Moon, Minhe Chae, Duckshin Park, Young-Chul Lee *
Reviewer 3:
I appreciated author’s efforts to improve the quality of the manuscript and their answer to my concerns, so I can now recommend publication of the manuscript. I am just still a little concerned about the use of the term “systemic” for botanical agents. Maybe authors should think about the possibility to use a different term, such as “coadjuvant”.
Response: Thank you for your appreciation. In fact, botanical agents (vitamin C and E, carotenoids, and phenolic and flavonoid compounds) are secondary metabolites produced by living organisms which play a crucial role in the growth and continuity of these organisms. These agents have strong anti-oxidant properties as well as can absorb UV rays. Moreover, as I mentioned in the previous response sheet, sun-protection systemic agents include several compounds that can behave as quenchers of free radicals or as stabilizers and can repair damaged DNA based on their anti-oxidant properties (Table S1) [1]. So, it can be said that the term “systemic” is proper to describe the typical properties of botanical agents for cosmetic applications instead of the term “coadjuvant”.
Table S1. Systemic photoprotective agents [1]
|
Agents |
Range |
Mechanisms |
|
Melanin |
UVB, UVA |
Optical filter and a stable free-radical that quenches other radicals |
|
Keratin |
UVB |
Optical filter |
|
Urocanic acid |
UVB |
Cis-trans-isomerization |
|
Carotenoids (-carotene) |
UVA + visible |
Quenchers for singlet oxygen |
|
Psoralens (8-methoxypsoralen and trimethylpsoralen) |
UVB |
Increased epidermal thickness and pigmentation
|
|
Antioxidants, ascorbic acid and -tocopherol (vitamin E) |
UVB |
Preferential oxidation |
|
Chloroquine |
UVB and UVA |
DNA and membrane stabilizer |
|
Oral antihistamines |
UVB |
Antihistamine |
|
Unsaturated fatty acids |
UVB |
Preferential oxidation |
|
Steroids (fluorinated and nonfiuorinated) |
UVB |
Anti-inflammatory |
|
Para-aminobenzoic acid |
UVB |
Optical filter |
|
Para-aminosalicylic acid |
UVB |
Optical filter |
|
Acetyl salicylic acid UVB |
UVB |
Lysosome stabilizer |
Reference
[1] Pathak, M.A. Sunscreens: Topical and systemic approaches for protection of human skin against harmful effects of solar radiation. J. Am. Acad. Dermatol. 1982, 7, 285-312.
